# Insecticidal Activities of *Sophora flavescens* Alt. towards Red Imported Fire Ants (*Solenopsis invicta* Buren)

**DOI:** 10.3390/toxins15020105

**Published:** 2023-01-24

**Authors:** Yongqing Tian, Zhixiang Zhang

**Affiliations:** Key Laboratory of Natural Pesticide and Chemical Biology, Ministry of Education, South China Agricultural University, Guangzhou 510642, China

**Keywords:** *Sophora flavescens*, *Solenopsis invicta*, matrine, sophocarpine

## Abstract

The red imported fire ant (*Solenopsis invicta*) is a worldwide invasive and dangerous insect that is controlled mainly by chemical insecticides. Plant-derived insecticidal compounds are generally better than synthetic insecticides for environmental compatibility and the biosafety of non-targets. The toxicity of the ethanol extract of *Sophora flavescens* roots against *S. invicta* was evaluated under laboratory conditions. The ethanol extract showed toxicity against minor and medium workers of *S. invicta* with 7-day LC_50_ values of 1426.25 and 2292.60 mg/L, respectively. By bioactivity-directed chromatographic separations using the minor worker as the test insect, two active compounds, matrine and sophocarpine, were isolated from the *S. flavescens* total alkaloids; their chemical structure was identified by ^13^C NMR data. Matrine showed toxicities against minor and medium workers with 7-day LC_50_ values of 46.77 and 71.49 mg/L, respectively, and for sophocarpine, 50.08 and 85.87 mg/L, respectively. The two compounds could substantially reduce the foraging response, food consumption, and aggregation of *S. invicta* workers at a sublethal concentration of 15 mg/L. The present research suggests that *S. flavescens* roots have potential as a natural control agent for red imported fire ants.

## 1. Introduction

The red imported fire ant (RIFA), *Solenopsis invicta* Buren, is one of the most notoriously known insects in the world [1]. This ant is native to South America and is a global invasive insect [2], and at present, it is distributed in many countries, including the United States, Mexico, Australia, New Zealand, China, Malaysia, Singapore, and the West Indies. This insect can cause harm to humans, native ecosystems, and economic activities worldwide [3]. In infested areas, RIFAs are commonly found in gardens, lawns, parks, school yards, golf courses, and on roadsides. This ant is very harmful to human health; they will attack a person who tramples or touches their nests by stings. The venom can cause intense burning, itching, blistering, or pustules, which will last for several weeks [4,5]. The broken or scratched pustules can harbor secondary bacterial infections. In worse cases, severe allergic reactions can result in anaphylactic shock and even death [6]. The ants reproduce rapidly in invaded areas and negatively impact native ecosystems and wildlife [7]. These ants can also inflict structural damage; for example, ant colonies in electrical circuitry can cause shorted air conditioners, and they may also inhabit telephone junction boxes, traffic and light control boxes, and transformers [3].

Management of RIFAs mostly relies on synthetic insecticide treatment in nests and surrounding areas. Although synthetic insecticides can be used to rapidly control RIFAs, their drawbacks cannot be ignored, such as environmental contamination, killing non-target organisms. Therefore, to solve these problems, sustainable approaches have been explored and integrated into pest management projects. Plant-derived compounds are one of the most attractive alternatives to synthetic insecticides [8], because they have many desirable attributes, such as an eco-friendly nature, availability, safety, acceptability, and minimal side effect on beneficial organisms, thus plants could be used as alternatives to synthetic chemical pesticides [9]. In the past, many scholars studied the activities of plants towards RIFAs, and these results were summarized by Chen and Oi [8]. According to this review, the active compounds of plant origin included potential fire ant bait active ingredients, contact toxins, repellants, and fumigants. Among these active compounds, two conventional botanical insecticides, pyrethrins, and rotenone, together with seven plant essential oils, namely d-limonene, clove oil, cotton seed oil, lemongrass oil, peppermint oil, pine oil, rosemary oil, and turpentine, have been registered for use against fire ants in the U.S.

In our studies on insecticidal plants, we found that a crude ethanol extract from the root of *Sophora flavescens* Alt., a herbaceous subshrub grown in China, was toxic to RIFA workers. In this research, we tested the bioactivities of total alkaloids from the root ethanol extract under laboratory conditions, then isolated two active compounds, matrine and sophocarpine, using bioactivity-guided separations. We also found that the alkaloids in the *S. flavescens* root had sublethal effects towards RIFAs, including a reduced foraging response, food consumption, and aggregation. Our study demonstrated that *S. flavescens* has potential as a biopesticide for controlling RIFAs.

## 2. Results

### 2.1. Structure Determination of Insecticidal Compounds

By bioactivity-directed separation, two active compounds were isolated and identified to be matrine and sophocarpine (Figure 1) based on the ^13^C NMR data (Figure 2 and Figure 3; Table 1) which were in agreement with the reported data [10,11].

### 2.2. Toxicities of Active Compounds

A bioassay showed that *S. flavescens* root ethanol extract possessed activities against minor and medium workers of RIFAs; the 7-day LC_50_ values were 1426.25 and 2292.60 mg/L (Table 2 and Table 3), respectively. Because alkaloids are the typical compounds contained in *S. flavescens*, the total alkaloids were separated from the ethanol extract using an acid solution and alkali isolation procedure; the subsequent bioassay showed the total alkaloids had higher activities than those of the ethanol extract; the 7-day LC_50_ values towards minor and medium RIFA workers were 214.67 and 371.49 mg/L, respectively, while the rest of the substance possessed no activity, which suggested that the active compounds were in the total alkaloids. Therefore, the total alkaloids were further separated, guided by the bioassay using the minor workers, and finally two active compounds, matrine and sophocarpine, were obtained. The bioactivities of matrine and sophocarpine were much higher than those of the root ethanol extract and total alkaloids; the 7-day LC_50_ values of matrine towards the minor and medium workers were 46.77 and 71.49 mg/L respectively, and for sophocarpine, 50.08 and 85.87 mg/L (Table 2 and Table 3), respectively.

### 2.3. Effects of Active Compounds on the Foraging Response of RIFAs

As shown in Figure 4, at sublethal concentrations, alkaloids in the *S. flavescens* root could affect the foraging response of RIFAs. After being treated with matrine, there were no significant differences in feeding rates among the three treatments during the first two days. On day 3, the feeding rate of 15 mg/L matrine was significantly lower than that of the control, but that of 7.5 mg/L matrine and the control did not differ significantly. Starting from day 4, the feeding rate of the three treatments became significantly different, and the differences continued to day 5. On day 5, the feeding rates of the control and 7.5 and 15 mg/L matrine were 96.67%, 82.35%, and 71.13%, respectively (Figure 4A). After being treated with sophocarpine, the pattern of the feeding rate was similar to that of matrine. On day 5, there were significant differences in the feeding rates among the control and 7.5 and 15 mg/L sophocarpine; the corresponding feeding rates were 96.67%, 84.73%, and 73.69%, respectively (Figure 4B). As for those of the total-alkaloids-treated groups, a slightly different pattern was observed. During the first three days, there were no significant differences in the feeding rates among the three treatments. On day 5, the feeding rates of 7.5 mg/L (88.34%) and 15 mg/L (84.17%) of total alkaloids were similar but significantly less than that of the control (96.67%) (Figure 4C).

### 2.4. Effects of Active Compounds on Food Consumption of RIFAs

The consumption of *T. molitor* (mealworm) by the ant workers treated with matrine, sophocarpine, and total alkaloids were examined. After being treated with matrine, there were no significant differences in the amount of *T. molitor* consumption among the three treatments on day 0 and day 3. On day 6, the consumption amount of workers treated with 15 mg/L matrine was significantly lower than that of the control, but there was no significant difference in the 7.5 mg/L treated group and the control. On day 9, significant differences could be seen among the three treated groups, and these differences continued to day 12. On day 12, the consumption rates of the control and the 7.5- and 15 mg/L matrine-treated groups were 0.14, 0.09, and 0.06 mg per ant, respectively (Figure 5A). After being treated with sophocarpine, the results were similar to those of matrine. On day 12, the consumption rates of the control and 7.5- and 15 mg/L sophocarpine treated groups were 0.16, 0.11, and 0.07 mg per ant, respectively (Figure 5B). As for the total alkaloids-treated groups, there were no significant differences among the three treatments during the first six days. On day 9, the *T. molitor* consumption of the 15 mg/L total-alkaloids-treated group was significantly less than that of the control, but the consumption of 7.5 mg/L total alkaloids and the control did not differ significantly. On day 12, there were significant differences among the control and the 7.5- and 15 mg/Ltotal alkaloids; the corresponding *T. molitor* consumption rates were 0.16, 0.13, and 0.09 mg per ant, respectively (Figure 5C).

### 2.5. Effects of Active Compounds on RIFA Aggregation

Alkaloids in the *S. flavescens* root could affect aggregation of RIFAs. After being treated with matrine (Figure 6A), no differences were recorded on day 0. From day 3 to day 6, the aggregation rate of the 15 mg/L matrine-treated group was significantly lower than the control, but the 7.5 mg/L matrine-treated group was not significantly different from the control. From day 9 to 12, the 7.5 mg/L matrine-treated group also showed significant differences to the control. During the whole experiment period, there were no differences between the 7.5 and 15 mg/L matrine-treated groups. The aggregation rate of workers treated with sophocarpine showed a similar tendency (Figure 6B). As for the total-alkaloids-treated groups, the 15 mg/L-treated group was significantly lower than the control from day 3 to day 12, but no significant differences occurred between the 7.5 mg/L-treated group and the control during the 12 days of evaluation (Figure 6C).

## 3. Discussion

The reported methods for controlling RIFAs included the use of chemical insecticides, such as hydramethylnon, indoxacarb [12], and fipronil [13], or biological control agents, such as entomopathogenic fungus *Beauveria bassiana* [14]. Plant-derived chemicals have attracted researchers’ attention because of their biodegradability, less harm to non-target organisms, and minimum effects on pest resistance [15]. Several plants have been studied that show activities towards RIFAs, such as *Syzygium aromaticum* [16], *Acorus calamus* [17], and *Cinnamomum osmophloeum* [18]. As far as is known, there has been no report on the use of *S. flavescens* for controlling this type of ant. This research showed that the ethanol extract of the *S. flavescens* root had activity towards RIFAs; the main active compounds were two alkaloids, matrine and sophocarpine. The two compounds could not only kill the RIFA workers, but also, at sublethal concentrations, they resulted in harmful chronic effects, including reduced foraging responses, consumption of *T. molitor*, and reduced aggregation. These chronic effects could affect their ability to attack other organisms and colony survival, which suggested another way of controlling this type of ant.

Approximately 300 compounds have been identified in *S. flavescens* [19,20], with most of them belonging to alkaloids. *S. flavescens* possesses diverse pharmacological properties including antianaphylaxis and antimicrobial, immunoregulatory, and cardioprotective activities [21]. These therapeutic effects of *S. flavescens* may be derived from complex interactions among its compounds. Inspired by the therapeutic effects, we conducted this research, and found the total alkaloids had bioactivity against RIFA workers, while the rest of the substance of the *S. flavescens* root ethanol extract showed no activity. Generally, the total alkaloids contained many compounds, but only those with high contents could be easily isolated, and it was difficult to isolate the minor constituents owing to their low contents. Thus, only two active compounds with high contents were isolated from the *S. flavescens* root in this work. According to the existing phytochemical research results, compounds in the same plant always have similar chemical structures and it has been suggested that they have similar bioactivities. Therefore, the minor alkaloids in the *S. flavescens* root may also have bioactivities against red fire ants, which needs further research.

The dried roots of *S. flavescens*, a traditional Chinese medicine [22], have been widely used in Korea, Japan, and China for the treatment of diarrhea, inflammation, abscesses, dysentery, and fever in East Asian countries [23]. Modern research has proven that compounds in this plant have clinical significance [24], such as treating diabetic retinopathy, tumour or cancer [25,26], and aerobic vaginitis [27]. These results show the *S. flavescens* compounds are safe for humans to some extent.

At present, the botanical pesticides are mainly maded from plant materials. *S. flavescens* can grow in more than ten provinces in China, and also distribute in Korea, Japan, India, etc., which make it possible to provide enough material for pesticide production. According to the above discussion, it can be preliminarily concluded that *S. flavescens* is a promising plant for controlling RIFAs.

## 4. Conclusions

In summary, this research demonstrated that the alkaloids in *S. flavescens* roots are lethal to RIFAs; this toxicity is mainly attributed to matrine and sophocarpine. At sublethal concentrations, the alkaloids could result in harmful chronic effects, including reduced foraging responses, food consumption, and aggregation, which could affect their ability to attack other organisms and colony survival. Thus, the alkaloids in *S. flavescens* roots have potential as natural insecticides for controlling RIFAs by either directly killing them at a high concentration or them reducing viability at a lower concentration. However, this work is only the beginning of developing *S. flavescens* alkaloids as an ideal control agent; the applicability and stability need to be further investigated.

## 5. Materials and Methods

### 5.1. Plant and Insects

Plant material: The roots of *S. flavescens* were purchased as a commercially available product from a standard drugstore, because it is a very common traditional Chinese medicine.

The origin and rearing of the RIFA workers: Workers of RIFAs were collected from the suburbs of Guangzhou city and were classified as minor (2.8–3.0 mm body length and 0.6–0.7 mm head width) and medium (3.5–3.7 mm body length and 0.8–0.9 mm head width) workers. The collected ants were fed with a mixture of 10% honey and ham sausage. A test tube (25 mm × 200 mm) partially filled with water and plugged with cotton was used as a water source. The ants were maintained in the laboratory at 25 ± 2 °C.

### 5.2. Chemicals and Reagents

Silica gel (60–100 and 100–200 mesh) and TLC precoated plates (GF_254_) were bought from Qingdao Marine Chemical Ltd., Qingdao, China. Sephadex LH-20 was bought from GE Healthcare Bio-Sciences AB, Uppsala, Sweden. CDCl_3_ was bought from Cambridge Isotope Laboratories, Inc., Andover, MA, USA. The organic solvents used in this work, ethanol, MeOH, EtOAc, CH_2_Cl_2_, and CHCl_3_, were all of analytical grade.

### 5.3. Extraction and Compound Isolation

Extraction and bioactivity-directed isolation of matrine and sophocarpine: The powdered dry roots of *S. flavescens* (6 kg) was extracted by the method of immersion with ethanol three times at room temperature; the ethanol solutions were combined and concentrated in vacuo. This crude ethanol extract (900.9 g) was subsequently dissolved in aqueous hydrochloric acid (to pH = 2) and filtered. The filtered aqueous acid layer was made alkaline to pH = 9 with NH_3_·H_2_O, then extracted three times with EtOAc to give 1.3 g total alkaloids. The total alkaloids and the rest of the substance were bioassayed using minor RIFA workers, which showed that the active compounds were in the total alkaloids and the rest of the substance possessed no activity. The total alkaloids were chromatographed over a silica gel column (100–200 mesh) and eluted with CH_2_Cl_2_–MeOH mixtures of increasing polarities (60:1–60:3), to yield seven fractions (I–VII) according to the TLC analysis. The activities of the fractions I–VII against *S. invicta* minor workers were tested at concentration of 500 mg/L. Fractions III and V, obtained on elution with CH_2_Cl_2_–MeOH (60:1–60:2), demonstrated strong activities towards *S. invicta* workers and caused 86.67% and 90.00% mean mortalities after 7-day treatment, respectively. The mean mortality of fraction IV was 38.89% and the rest of the fractions did not display any activity. Fractions III and V were respectively separated by a Sephadex LH-20 column, eluted with CHCl_3_–MeOH (1:1), to give two active insecticidal compounds, sophocarpine (70.8 mg) and matrine (90.2 mg).

The ^1^H (600 MHz) and ^13^C (150 MHz) NMR spectra of the isolated compounds were recorded in CDCl_3_ on a Bruker AV-600 instrument using TMS as an internal reference. For column chromatography, silica gel and Sephadex LH-20 were used. TLC was performed on precoated plates with detection under a fluorescent (λ = 254 nm) light, in iodine vapor and using Dragendroff’s reagent.

### 5.4. Toxicities of Active Compounds

The activity of the crude ethanol extract of the *S. flavescens* root, the total alkaloids, and the active compounds against RIFA workers were evaluated using the “water tube” method [28] with slight modifications. The water source was a test tube (10 mm × 30 mm) filled approximately two-thirds full with solution and tightly plugged with a saturated cotton. The solution in the test tube contained the abovementioned crude ethanol extract, total alkaloids, and the active compounds dissolved in a DMSO/water (1:99, *v*/*v*) mixture. The water test tube and worker ants were placed in a disposable plastic cup (top/bottom/height: 62 mm/40 mm/60 mm) whose vertical wall was precoated with a fluon emulsion and allowed to dry for 24 h to prevent the ants from escaping. Each treatment was replicated three times, and each replicate included 30 worker ants. The test tube in the control group was filled with a DMSO/water (1:99, *v*/*v*) mixture. The treated workers were maintained at 24 °C to 26 °C and 60% to 80% relative humidity and were fed with ham sausage. The average mortalities of the three replications at each concentration were recorded on the 7th day after treatment and were corrected by Abbott’s formula [29]. The median of the lethal concentration value (LC_50_), which was defined as the concentration causing 50% mortality, was determined using probit analysis.

For activity-directed separation, the mortalities of fractions from the subsequent separation procedure towards RIFA workers were assessed at the concentration of 500 or 1000 mg/L using the method described above.

### 5.5. Effects of Active Compounds on the Foraging Response of RIFAs

RIFA workers were treated in the same manner as those mentioned above at sublethal concentrations, 7.5 and 15 mg/L. Minor workers, about 3 mm in length [18,30], were used for recording the foraging responses. A total of 30 workers were tested for each treatment and the control group, and each treatment had three replicates. The foraging responses were observed at the 24th, 48th, 72nd, 96th, and 120th hour after treatment. Workers were considered to possess foraging abilities if they could feed on the fresh sausage within 10 min after the fresh sausage was added. The foraging response of the RIFAs was evaluated by the feeding rate; the equation was as follows: feeding rate (%) = (number of ants possessing a feeding ability/number of alive workers) × 100.

### 5.6. Effects of Active Compounds on the Food Consumption of RIFAs

RIFA minor workers were treated in the same manner as the above method at sublethal concentrations. On the 0, 3rd, 6th, 9th, and 12th day, *T. molitors* was weighed and then added to the plastic cup as ant food. After 24 h of feeding, the *T. molitors* was removed from the cup and weighed to determine the consumption of *T. molitor* (mg *T. molitor* per ant), then the average consumption was calculated. A total of 60 minor workers were tested for each treatment and the control group, and each treatment had three replicates. The consumption of *T. molitor* was calculated using the following equation: consumption of *T. molitor* (%) = (weight loss of *T. molitor*/number of alive ants) × 100.

### 5.7. Effects of Active Compounds on the Aggregating Rate of RIFAs

RIFA workers were treated in the same manner as the above method at sublethal concentrations. A total of 30 minor workers were tested for each treatment and the control group, and all treatments were replicated thrice. Aggregation was measured every 3 days up to 12 days of treatment; aggregation was defined as the gathering of more than five workers. The aggregation rate was calculated using the following equation: aggregation rate (%) = (number of aggregated workers/number of alive workers) × 100.

### 5.8. Data Analysis

The collected data from the aforementioned experiments were subjected to one-way analysis of variance (ANOVA) using SPSS Statistics, Version 17.0, 2009 (International Business Machines Corporation, Armonk, NY, USA). If significant differences occurred among the treatments, the means were separated by Tukey’s honestly significant difference (HSD) test at the *p* < 0.05 level. The means were presented in graphs with standard error bars which were drawn using OriginPro (Version 2021, OriginLab Corporation, Northampton, MA, USA).

## Figures and Tables

**Figure 1 toxins-15-00105-f001:**
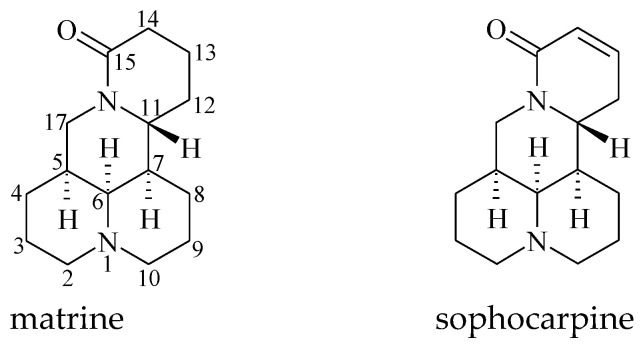
Chemical structures of matrine and sophocarpine.

**Figure 2 toxins-15-00105-f002:**
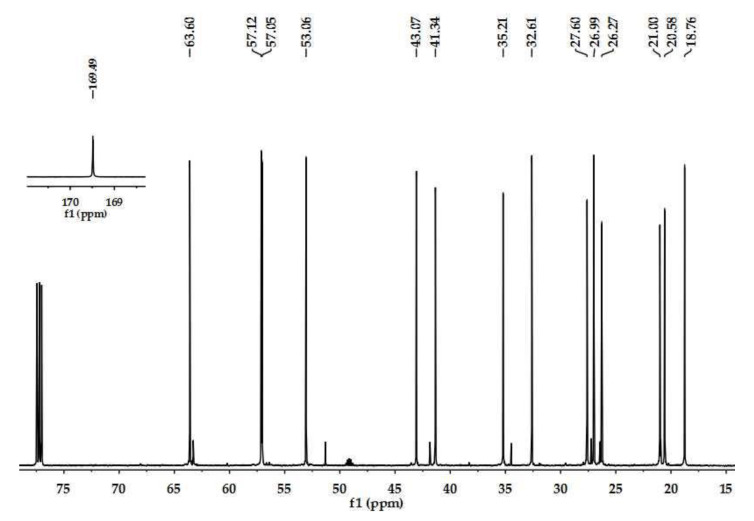
^13^C NMR spectrum of matrine (in CDCl_3_).

**Figure 3 toxins-15-00105-f003:**
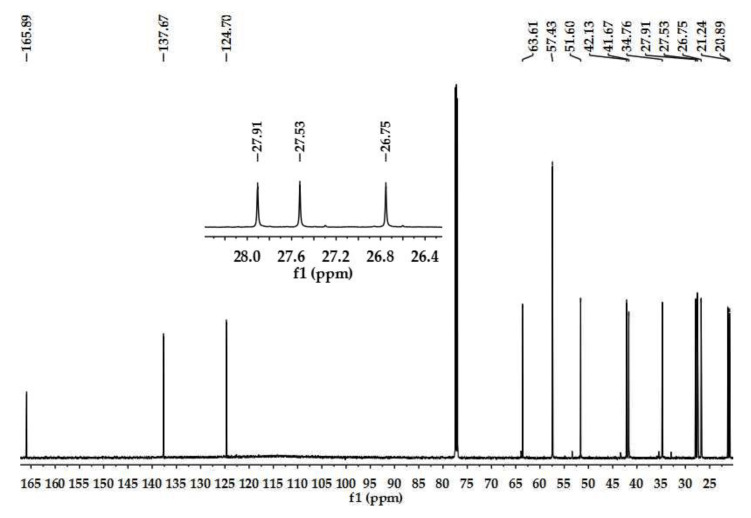
^13^C NMR spectrum of sophocarpine (in CDCl_3_).

**Figure 4 toxins-15-00105-f004:**
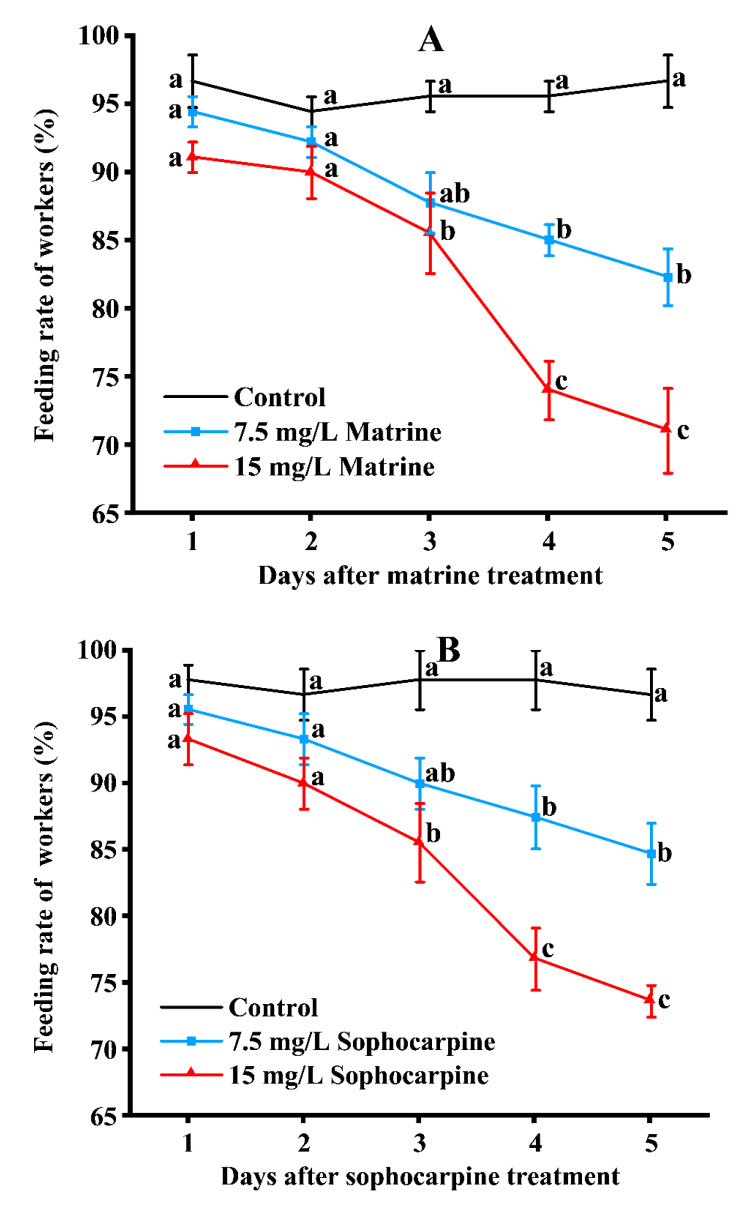
Foraging response of RIFA minor workers after being treated with matrine (**A**), sophocarpine (**B**), and total alkaloids (**C**). Data are presented as mean ± S.E. Different letters on each sampling day indicate significant differences per parameter among treatments at the *p* < 0.05 level based on Tukey’s HSD test (*n* = 3).

**Figure 5 toxins-15-00105-f005:**
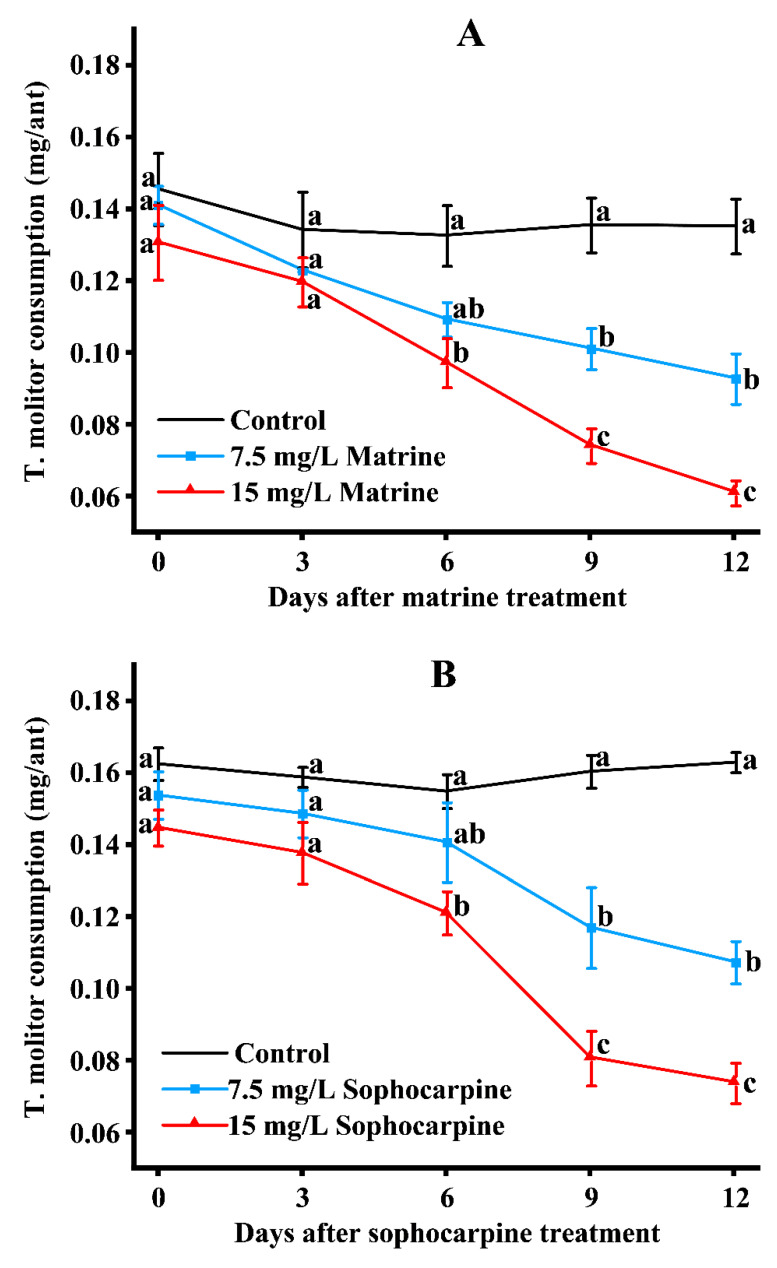
The amount of *T. molitor* consumed by RIFA minor workers after treatment with matrine (**A**), sophocarpine (**B**), and total alkaloids (**C**). Data are presented as mean ± S.E. Different letters on each sampling day indicate significant differences per parameter among treatments at the *p* < 0.05 level based on Tukey’s HSD test (*n* = 3).

**Figure 6 toxins-15-00105-f006:**
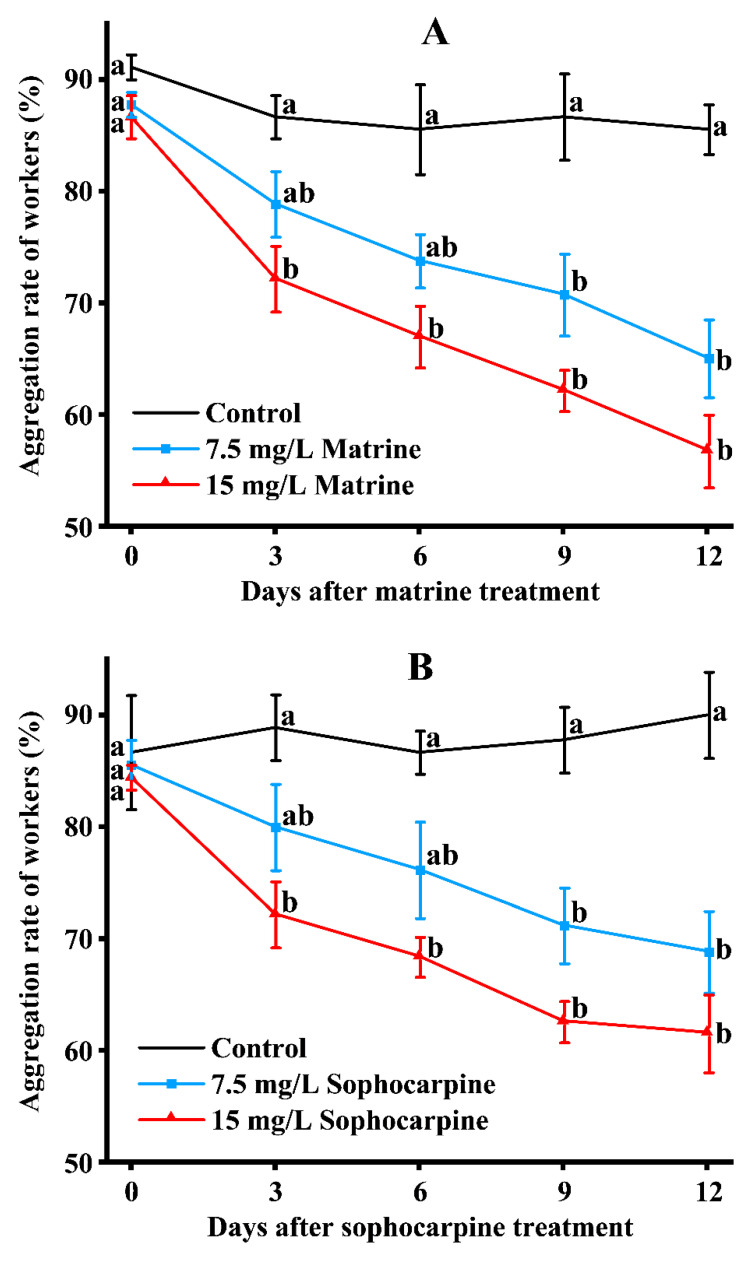
The aggregating rate of RIFA minor workers fed with matrine (**A**), sophocarpine (**B**), and total alkaloids (**C**). Data are presented as mean ± S.E. Different letters on each sampling day indicate significant differences per parameter among treatments at the *p* < 0.05 level based on Tukey’s HSD test (*n* = 3).

**Table 1 toxins-15-00105-t001:** ^13^C NMR assignments of matrine and sophocarpine in CDCl_3_.

Assignment	Matrine (*δ*)		Sophocarpine (*δ*)	
	Observed	Reference [10]	Observed	Reference [11]
2	57.12	57.6	57.43	57.3
3	21.00	21.4	21.24	21.1
4	27.60	27.9	27.91	27.8
5	35.21	35.6	34.76	34.7
6	63.60	64.0	63.61	63.5
7	43.07	43.4	41.67	41.6
8	26.27	26.6	26.75	26.6
9	20.58	21.0	20.89	20.8
10	57.05	57.4	57.43	57.3
11	53.06	53.4	51.60	51.5
12	26.99	27.4	27.53	27.4
13	18.76	19.2	137.67	137.3
14	32.61	33.0	124.70	124.5
15	169.49	169.7	165.89	165.4
17	41.34	41.7	42.13	41.9

**Table 2 toxins-15-00105-t002:** LC_50_ values (mg/L) of the *S. flavescens* ethanol extract, total alkaloids, matrine, and sophocarpine against RIFA minor workers.

Treatment	Regression Equations	7-Day LC_50_ (95% Confidence Interval)	Correlation Coefficient (R^2^)
*S. flavescens* ethanol extract	Y = −0.3859 + 1.7075 x	1426.25 (1201.21~1693.45)	0.9838
Total alkaloids	Y = 1.7000 + 1.4152 x	214.67 (169.25~272.27)	0.9982
Matrine	Y = 1.9605 + 1.8201 x	46.77 (39.74~55.04)	0.9953
Sophocarpine	Y = 2.1387 + 1.6835 x	50.08 (42.12~59.54)	0.9969
Fipronil (control)	Y = 5.1185 + 2.2215 x	0.88 (0.76~1.03) ^1^	0.9919

^1^ 1-day LC_50_ value.

**Table 3 toxins-15-00105-t003:** LC_50_ values (mg/L) of the *S. flavescens* ethanol extract, total alkaloids, matrine, and sophocarpine against RIFA medium workers.

Treatment	Regression Equation	7-Day LC_50_ (95% Confidence Interval)	Correlation Coefficient (R^2^)
*S. flavescens* ethanol extract	Y = 1.2697 + 1.1101 x	2292.60 (1740.50~3019.84)	0.9965
Total alkaloids	Y = 1.5613 + 1.3380 x	371.49 (297.38~464.07)	0.9900
Matrine	Y = 1.7937 + 1.7292 x	71.49 (60.28~84.77)	0.9952
Sophocarpine	Y = 1.6500+ 1.7323 x	85.87 (72.18~102.15)	0.9858
Fipronil (control)	Y = 4.7842 + 1.6863 x	1.34 (1.13~1.60) ^1^	0.9920

^1^ 1-day LC_50_ value.

## Data Availability

Data are contained within the article.

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
