# Peer review of "Insecticidal Activities of Sophora flavescens Alt. towards Red Imported Fire Ants (Solenopsis invicta Buren)"

_toxins, 2023, doi:10.3390/toxins15020105_

Round 1
Reviewer 1 Report
The introduction could be improved with a more detailed description of the ongoing studies/results (and if they are published, they must be refferenced).
In the methodology, the rename of subsection and adding of one other should be considered (subsection 2 is not chemicals, is extraction and compound isolation; and a subsection oh chemicals and reagents is missing)
A conclusion section must be added.
Reviewer 2 Report
The Authors present a straightforward and concise manuscript describing the insecticidal activities of two alkaloids, matrine and sophocarpine, isolated from Sophora roots against the red imported fire ant (RIFA, Solenopis invicta), which is a dangerous and invasive species. Although the toxic effect of these alkaloids is, arguably, not robust particularly within 24-48 hours, the authors provide convincing evidence, supported by statistical analyses, that these chemical or modifications thereof, potentially have applied value in the control of RIFA, perhaps in integrated pest management control programs.
The Reviewer has no concerns about the scientific design, data acquisitions, statistical analyses, presentation of figures, and conclusion.
Minor comments:
1. Have these alkaloids been tested against animal models systems, including in in vivo studies or cell lines?
2. Minor editing of the manuscript is required, particularly in the abstract section, and Introduction section
Examples:
Abstract:
a. "nowadays it was contyrelled mainly..." consider: "that is controlled mainly by chemical insecticides."
b. Delete or rewrite the second sentence; the term "better" is arguable, and shows bias.
c. Last sentence: "natural control agent"
Introduction:
Here are some suggestions for editing:
" ..., and at present is distributed..."
"...can cause harm to humans, natgive ecosystems..."
"In infested areas, RIFA is..."
"These ants can also inflict structural damage, for example,
"Although synthetic insecticides can be used to rapidly control RIFA, their drawbacks..."
"Therefore, to solve these problems..."
Paragraph 2: Clarfiy what is meant by "and the rest had been assessed for their bioactivities"
"Was toxic to" ...delete "toward"
Reviewer 3 Report
The manuscript provided is dedicated to the study of red ants and the screening of alkaloids, which can be used as a potential pesticides. The article provides interesting data, but has some flaws:
1. The comparison with "classical" synthetic pesticides should be add. It is difficult to say, how strong the effects are.
2. The results for the other fractions of alkaloids should be added.
3. To say that the substances might be used as biopesticides, some test for the biosafety should be added. Are they toxic for vertebrates?
4. The effects of sublethal concentrations seems to be just a result of toxicity. It is not clear, why it is important? Can it helps to explain the mechanisms?
Overall, the only important result of the manuscript is the identification of the toxic substances in the extract known to be toxic. I would suggest adding the mode of action for the substances, or the tests with other animals.
